# A Novel CNN pooling layer for breast cancer segmentation and classification from thermograms

**Esraa A. Mohamed**[1], **Tarek Gaber**[2,3]*, **Omar Karam**[4], **Essam A. Rashed**[1,5]

**1** Faculty of Science, Department of Mathematics, Suez Canal University, Ismailia, Egypt, **2** Faculty of Computers and Informatics, Suez Canal University, Ismailia, Egypt, **3** School of Science, Engineering and Environment University of Salford, Manchester, United Kingdom, **4** Faculty of Informatics and Computer Science, British University in Egypt (BUE), Cairo, Egypt, **5** Graduate School of Information Science, University of Hyogo, Kobe, Japan

* t.m.a.gaber@salford.ac.uk

**Data Availability Statement:** All relevant data are within the paper and its Supporting Information files.

**Funding:** The author(s) received no specific funding for this work.

## Abstract

Breast cancer is the second most frequent cancer worldwide, following lung cancer and the fifth leading cause of cancer death and a major cause of cancer death among women. In recent years, convolutional neural networks (CNNs) have been successfully applied for the diagnosis of breast cancer using different imaging modalities. Pooling is a main data processing step in CNN that decreases the feature maps' dimensionality without losing major patterns. However, the effect of pooling layer was not studied efficiently in literature. In this paper, we propose a novel design for the pooling layer called vector pooling block (VPB) for the CCN algorithm. The proposed VPB consists of two data pathways, which focus on extracting features along horizontal and vertical orientations. The VPB makes the CNNs able to collect both global and local features by including long and narrow pooling kernels, which is different from the traditional pooling layer, that gathers features from a fixed square kernel. Based on the novel VPB, we proposed a new pooling module called AVG-MAX VPB. It can collect informative features by using two types of pooling techniques, maximum and average pooling. The VPB and the AVG-MAX VPB are plugged into the backbone CNNs networks, such as U-Net, AlexNet, ResNet18 and GoogleNet, to show the advantages in segmentation and classification tasks associated with breast cancer diagnosis from thermograms. The proposed pooling layer was evaluated using a benchmark thermogram database (DMR-IR) and its results compared with U-Net results which was used as base results. The U-Net results were as follows: global accuracy = 96.6%, mean accuracy = 96.5%, mean IoU = 92.07%, and mean BF score = 78.34%. The VBP-based results were as follows: global accuracy = 98.3%, mean accuracy = 97.9%, mean IoU = 95.87%, and mean BF score = 88.68% while the AVG-MAX VPB-based results were as follows: global accuracy = 99.2%, mean accuracy = 98.97%, mean IoU = 98.03%, and mean BF score = 94.29%. Other network architectures also demonstrate superior improvement considering the use of VPB and AVG-MAX VPB.

**Competing interests:** The authors have declared that no competing interests exist.

**Abbreviations:** CNN, Convolutional Neural Networks; CAD, Computer-Aided Detection; VPB, Vector pooling block; AVG-MAX VPB, AVG-MAX vector pooling block; ADAM, Adaptive Moment Estimation; $T_P$, True Positive; $T_N$, True Negative; $F_P$, False Positive; $F_N$, False Negative; FCN, Fully Convolutional Network; RELU, Rectified Linear Activation Function; IoU, Intersection over union; BF Score, Boundary f1 score.

## 1. Introduction

Breast cancer is the second most frequent cancer in the world, following lung cancer, the fifth leading cause of cancer death and the major cause of cancer death among women [1]. Breast cancer can affect both men and women, however women are diagnosed with the disease 100 times more frequently than men [1]. The most important challenges in breast cancer detection process are accurate segmentation of the breast area and classification of the breast tissue, which play an important role in image guiding surgery, radiological treatment, and clinical computer-assisted diagnosis [2]. Several breast imaging modalities are currently being used for early detection of breast cancer such as ultrasound [3], mammography [4], MRI [5], thermography [6,7], etc. Computer aided detection (CAD) system is used for the diagnosis of breast cancer. This diagnosis contains several methods and techniques, including image processing, machine learning [7], data analysis, artificial intelligence [8], and deep learning [6,9,10].

Deep learning is a machine learning technology that uses multilayer convolutional neural networks (CNNs) [11]. It has a significant effect in fields associated to medical imaging such as brain tumor detection as in [12] which proposes a fully automated design to classify brain tumors, COVID-19 as in [13,14] which propose a deep learning and explainable AI technique for the diagnosis and classification of COVID-19 using chest X-ray images and [14] which proposed a CNN-LSTM and improved max value features optimization framework for COVID-19 classification to address the issue of multisource fusion and redundant features, lung cancer [15], which developed and validated a deep learning-based model using the segmentation method and assessed its ability to detect lung cancer on chest radiographs, etc.

In early stages of breast cancer, the detectability of abnormal tissues is challenging using standard approaches such as mammography as it localized in small region and usually presented in texture similar to surrounding normal breast tissues [16]. However, in thermography, the detectability is based on the change in body temperature which open additional potentials for early detection using different physical features. Machine learning and deep learning approaches lead to improve the performance of detectability due to the ability to recognize the image features without the need of feature engineering. Therefore, CNNs have been successfully applied for the diagnosis of breast cancer in recent years [6,10,17]. Investigated tasks include extracting the breast region from other parts of the body [6,18], segmentation of the breast cancer tumor lesion [19,20], and classification of the breast tissue whether it is normal or abnormal [6,10,21,22].

A CNN architecture is typically composed of different layers. Most common layers are convolution, Rectified Linear Activation Function (ReLU), pooling, fully connected, and dropout [6]. Pooling is a main step in CNNs that decreases the feature maps' dimensionality. This is done by combining a set of values into a smaller set of values. It turns the joint feature representation into useful information by keeping only the most important data and discarding the rest. Pooling operators can provide a type of spatial transformation invariance in addition to reducing the computational complexity for upper layers by eliminating some connections between convolutional layers. Pooling layer performs down-sampling on the feature maps from the previous layer, resulting in new feature maps with a reduced resolution. It has two major purposes: 1) decreasing the number of parameters or weights, thus reducing the computational cost. 2) controlling overfitting, which is a well-known drawback of CNN. A perfect pooling method is supposed to extract only valuable information and discard inappropriate features. Despite these advantages of the pooling layer, it still has some drawbacks as losing features and reducing the spatial resolution which could affect the accuracy of the classification systems [23].

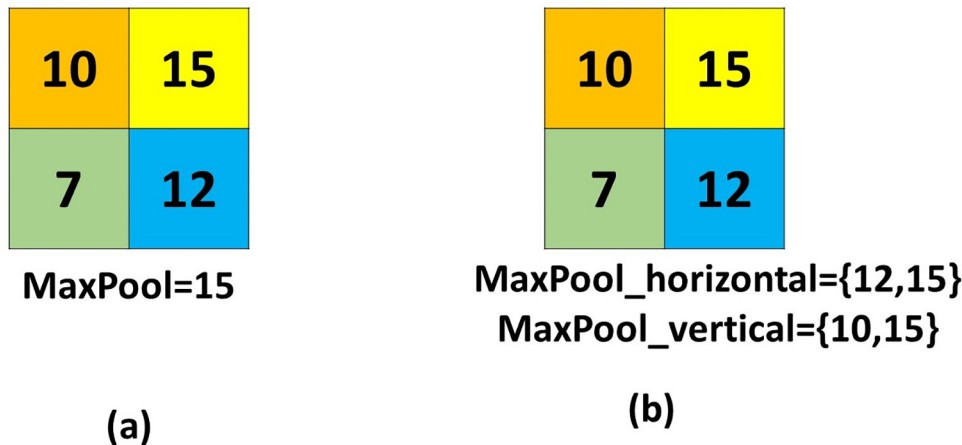

**Fig 1. The difference between traditional pooling method and the proposed pooling method.**

In this paper, we present a novel design for the pooling layer called vector pooling block (VPB). The VPB consists of two pathways, which focus on extracting features along horizontal and vertical directions. Pooling operation in different orientations has some advantages: 1) it uses a long kernel size in one dimension to extract more features with isolated regions. 2) it uses a small kernel size in the other direction, which is helpful in extracting local features. So, the VPB makes the CNNs able to observe both global and local features by including long and narrow pooling kernels, which is different from the traditional pooling layer, that gathers features from a fixed square kernel. The proposed pooling method has the ability to collect more features that are ignored by the traditional pooling method. Fig 1 illustrates the difference between the usage of traditional pooling method and the proposed method. Fig 2 illustrates another disadvantage of the traditional pooling method. From Fig 2, it can be noted that all the outputs of the traditional MaxPooling method from these matrices are the same despite that the matrices are different, but with using the proposed pooling method it will lead to different outputs that address the difference in a more efficient way.

Based on VPB, we proposed a pooling module called AVG-MAX VPB. It can collect informative features by using two types of pooling techniques, Max-pooling and average pooling,

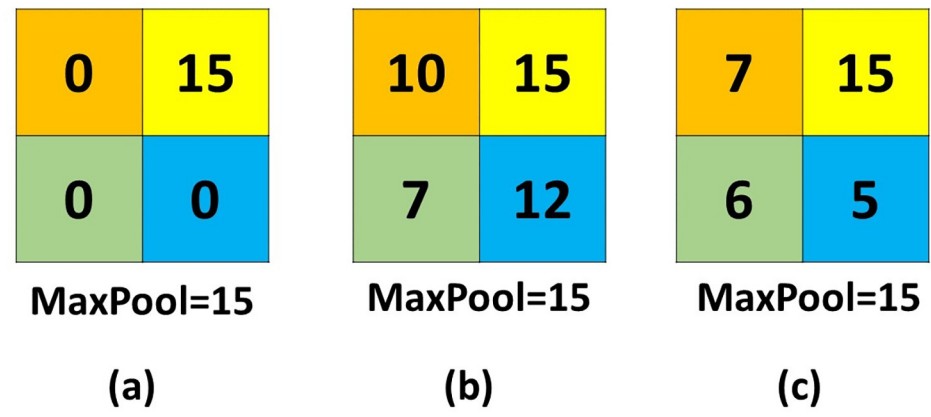

**Fig 2. MaxPooling of three different matrices is the same.**

with the concept of VPB. It can be used with different kernel shapes and can be incorporated with any CNNs used for segmentation or classification tasks.

The major contributions of this paper are as following:

1. Proposing a novel design for the pooling layer called VPB, which focus on extracting features along horizontal and vertical orientations and makes the CNNs able to collect both global and local features by including long and narrow pooling kernels

2. Proposing another new pooling block based on the VPB called AVG-MAX VPB, which creates a pooling block using average pooling and max pooling based on the concept of vector pooling.

3. Proposing an enhanced CNN (VPB-CNN) by embedding the proposed pooling models above and then used it in the semantic segmentation and classification of thermography breast cancer problem.

4. Conducing a thorough evaluation of the enhanced CNN (VPB-CNN) including the standard networks such as U-Net, AlexNet, ResNet18 and GoogleNet. This showed that the VPB-CNN outperformed these standard ones.

The structure of the paper is as follows. Section 2 explains the related work and Section 3 explains the proposed method. Section 4 contains the experimental results. Finally, the paper is discussed in Section 5 and concluded in Section 6.

## 2. Related work

In recent years, CNNs have become a very useful tool used for breast cancer segmentation and classification from thermal images due to their ability of automatically feature extraction from input data and the availability of software libraries that implement their functionality. One of the most effective state-of-the-art networks for semantic image segmentation is Fully Convolutional Network (FCN) [24]. Tayel and Elbagoury [25] used FCN-AlexNet as an end-to-end network for fully automated breast area segmentation form thermal images. They obtained 96.4% of accuracy, 97.5% of sensitivity and 97.8% of specificity. U-Net is improved and extended from FCN [26]. It can be used for classification such as [27] which uses U-Net CNN for the classification of brain tumor. It is widely applied to several medical image segmentation tasks [28] such as lung [29], skin lesions [30], etc. Also, it is used for breast area segmentation from thermal images. Baffa et al. [31] used U-Net for breast segmentation from thermograms and compared the segmentation method with state-of-the-art and machine learning segmentation methods. They reached Intersection-Over-Union (IoU) = 94.38%. De Carvalho et al. [32] used U-Net for breast area segmentation from frontal and lateral view of thermal images. They achieved an accuracy of 98.24% over the frontal view, and 93.6% over the lateral view. Mohamed et al. [6] used U-Net to automatically extract and isolate the breast area from the rest of body in thermograms. The segmentation method helped them in the classification process as they reached accuracy = 99.33%, sensitivity = 100% and specificity = 98.67%. However, the improvement of the segmentation performance with U-Net, but it is still having some drawbacks such as the pooling operation which may lose some important features that can improve the segmentation accuracy. Also, the high computational complexity produced from the continuous stacked convolutional layers which used to enhance the capability of U-Net for feature extraction [33]. To solve these drawbacks of U-Net, several scientific research worked on the improvement of U-Net for different medical image segmentation problems. Gu et al. [34] proposed a comprehensive attention-based convolutional neural network (CA-Net) to improve the performance and explainability of medical image segmentation. They integrated

their proposed method into most semantic segmentation networks such as U-Net. Oktay et al. [35] proposed a novel attention gate (AG) model which can be applied to medical image segmentation. They proved that AGs can be easily plugged into common CNN models such as U-Net with minimizing the computational time and increasing the model sensitivity and accuracy. Baccouche *et al*. [36] presented a model, called Connected-UNets, which connects two U-Net models by the usage of additional modified skip connections. they emphasized the contextual information by integrating Atrous Spatial Pyramid Pooling (ASPP) in the two conventional U-Net models. Also, they applied the proposed model on the Attention U-Net and the Residual U-Net.

As previously mentioned, several research have been conducted to investigate the problem of breast cancer classification from thermograms by using CNNs due to the ability of CNNs to extract complex features automatically. Mohamed *et al*. [6] presented a deep learning model based on two-class CNN, which is trained from scratch and used for the classification of normal and abnormal breast tissue from thermal images. Sánchez-Cauce *et al*. [37] proposed a model for early detection of breast cancer by combining the lateral views and the front view of the thermal images to enhance the performance of the classification model. Also, they built a multi-input classification model which exploits the benefits of CNNs for image analysis. They reached a 97% accuracy with a specificity of 100% and a sensitivity of 83%. Aidossov et al. proposed an efficient CNN model for binary classification of breast thermograms. The most important improvement of their work is the use of breast thermograms with multi-view images from a multicenter database without preprocessing for the binary classification. the model achieved an accuracy of 80.77%, sensitivity of 44.44% and the specificity of 100%. Alqhtani [38] proposed a novel layer-based Convolutional Neural Network (BreastCNN) for breast cancer detection and classification. The proposed technique worked in five different layers and utilized various types of filters. Accuracy of 99.7% is reached. Gomez *et al*. [39] investigate the effect of data preprocessing, data augmentation and the size of database in comparison to a set of proposed CNN models. Additionally, they developed a CNN hyper-parameter fine-tuning optimization model by using a tree Parzen estimator. They attained an F1-score of 92% and an accuracy of 92%.

Pooling is a main step in CNN that decreases the feature maps' dimensionality, but it has some drawbacks such as losing features and reducing the spatial resolution. Therefore, scientific researchers have been working on developing it to overcome these defects. Yu *et al*. [23] proposed a feature pooling method called mixed pooling to regularize CNNs, which replaces the traditional pooling operations with a stochastic method by randomly using the max pooling and the average pooling procedures. Their proposed pooling method can solve the overfitting problem faced by CNN generation. Lee *et al*. [40] investigate different approaches to enable the pooling layer to learn and adjust to complex and variable patterns. They presented two primary directions for the pooling function 1) learning the pooling function by combining the max pooling and the average pooling with two strategies mixed max-average pooling and gated max-average pooling, and 2) learning a pooling function in the form of a tree-structured fusion of pooling filters that are themselves learned. However, the proposed pooling operations enhance the performance of CNNs, but they increased the computational complexity and the number of parameters of the model. Tong and Tanaka [41] proposed a pooling method called hybrid pooling method to enhance the generalization ability of CNNs. The hybrid pooling method which selects the max pooling or the average pooling in each pooling layer stochastically. The probability for selecting the pooling model can be controlled for each convolutional layer. The experimental results with benchmark datasets show that the hybrid pooling increased the generalization capability of CNNs. Hssayni and Ettaouil [42] proposed a pooling method called $l^{1/2}$ pooling to enhance the generalization ability of Deep Convolutional Neural

Networks (DCNNs). Also, they combined their proposed method with additional regularization techniques like dropout and batch normalization, so they were able to achieve the state-of-the-art classification performance with moderate parameters.

From the discussed related work about the developed pooling layer, it could be remarked that the prior work has some limitations such as:

1. All related work developed the pooling layer for classification process only and they didn't work on the segmentation process.

2. The developed pooling methods in the related work has been evaluated by only calculating the accuracy metric only. However, the high accuracy rate of a model does not ensure its ability to distinguish different classes equally if the dataset is unbalanced [43].

3. The developed pooling methods in the related work hasn't been tested on a breast cancer dataset.

## 3. Proposed method

### 3.1. Vector pooling block

Let $I_{input}$ be an input layer of size $C{\times}H{\times}W$, where $C$ is the number of channels, $H$ is the height and $W$ is the width. First, $I_{input}$ is fed into two parallel paths, horizontal pooling layer and vertical pooling layer. Then, the output of each pooling path is followed by $1x1$ convolutional layer with number of kernels equal to the number of kernels of the previous layer. $1x1$ convolutional layer is used to extract more abundant features. Each $1x1$ convolution layer in the vector pooling block is followed by a ReLU layer for more stable performance and faster convergence. There are several methods to combine the extracted features from the two paths and preserve the dimensionality reduction of the pooling layer such as element-by-element summation or inner product between the vectors of the extracted features. To increase the efficiency of the vector pooling block, element-by-element summation is used to combine the extracted feature vectors. The element-by-element summation is followed by a RELU layer for faster convergence. The vector pooling block is shown in Fig 3 and can be expressed as following:

$$y_{vertical} = RELU\left(Conv_{1x1}(Pool_{1xN}(I_{Input}))\right) \tag{1}$$

$$y_{horizontal} = RELU\left(Conv_{1X1}(Pool_{NX1}(I_{Input}))\right) \tag{2}$$

$$y_{output} = RELU\left(y_{vertical} \oplus y_{horizontal}\right) \tag{3}$$

where $\oplus$ is the element-by-element summation.

### 3.2. AVG-MAX VPB

The most two conventional pooling methods used in CNNs are max-pooling and average pooling [23]. The max-pooling selects the maximum value in the pooling region [23]. The average pooling calculates the arithmetic mean of the elements in the pooling region [44]. Fig 4 shows an example of calculating max-pooling and average pooling. From Fig 4, the left side represent the input matrix of size $4x4$ and the right side is the calculation of the max-pooling and the average pooling with filter size $2x2$ and stride 2. Each cell color in right side represents the calculation of the max-pooling and the average pooling of this color from the input matrix.

AVG-MAX VPB creates a pooling block that can collect informative features by using two types of pooling techniques, Max-pooling and average pooling, with the concept of VPB.

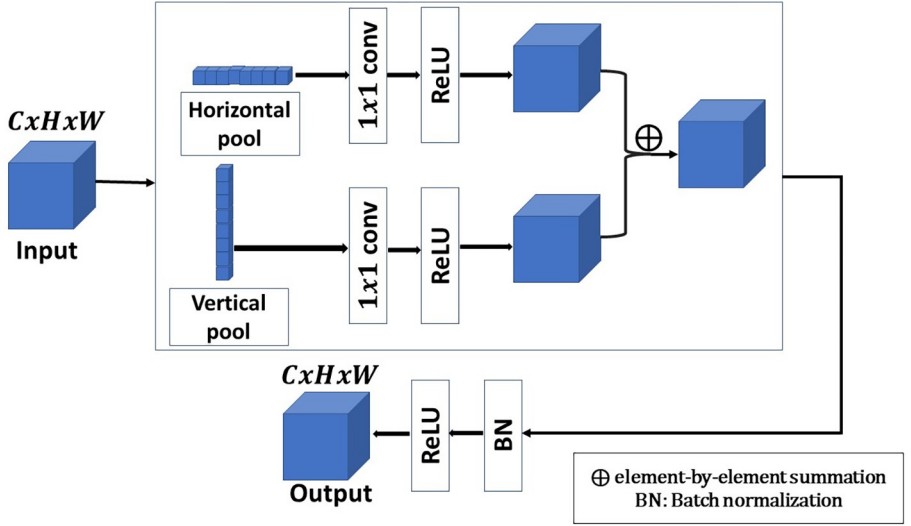

**Fig 3. Vector pooling block.**

AVG-MAX VPB is shown in Fig 5 and can be expressed as following:

$$y_{MAX-vertical} = ReLU\left(Conv_{1x1}(MaxPool_{1xN}(I_{Input}))\right) \qquad (4)$$

$$y_{MAX-horizontal} = ReLU\left(Conv_{1X1}(MaxPool_{NX1}(I_{Input}))\right) \qquad (5)$$

$$y_{AVG-vertical} = ReLU\left(Conv_{1x1}(Average\_Pool_{1xN}(I_{Input}))\right) \qquad (6)$$

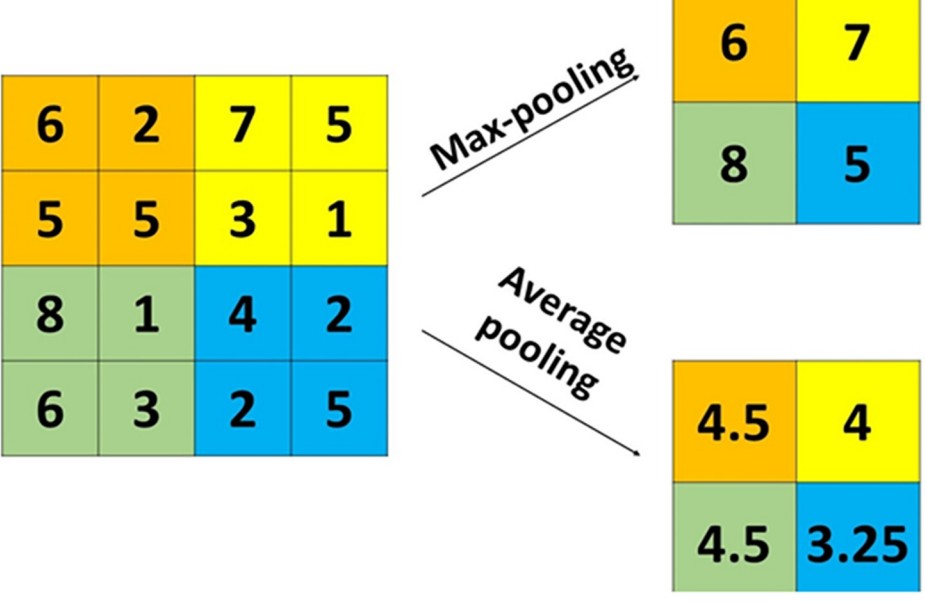

**Fig 4. Example of calculating max-pooling and average pooling with filter of size 2X2 and stride 2.**

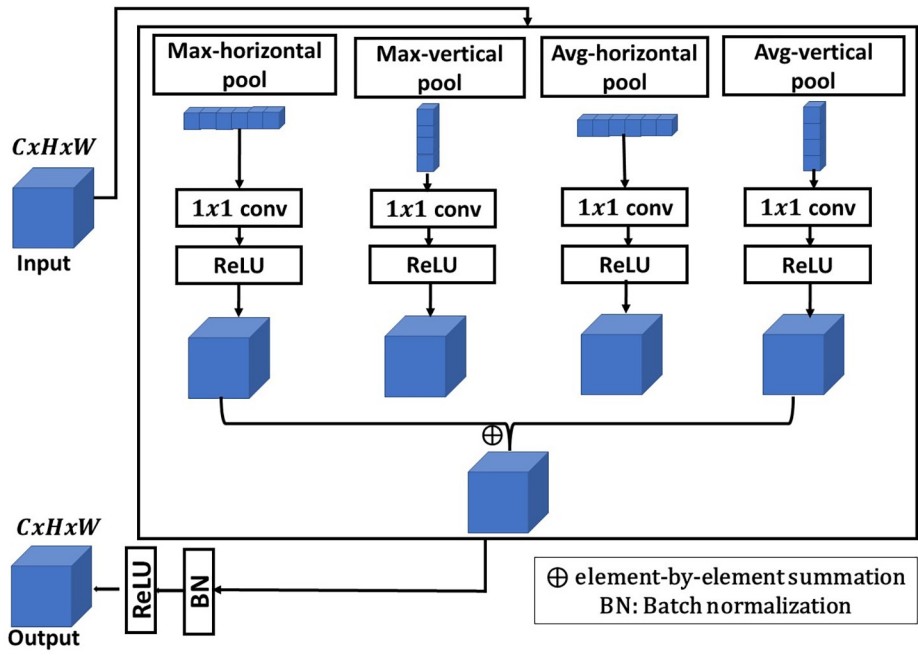

**Fig 5.**

$$y_{AVG-horizontal} = ReLU\left(Conv_{1X1}\left(Average\_Pool_{NX1}\left(I_{Input}\right)\right)\right) \tag{7}$$

$$y_{output} = RELU\left(y_{MAX-vertical} \oplus y_{MAX-horizontal} \oplus y_{AVG-vertical} \oplus y_{AVG-horizontal}\right) \tag{8}$$

From Fig 5, we can note that the input is fed into four parallel paths, Max-horizontal pooling layer, Max-vertical pooling layer, average horizontal pooling layer and average vertical pooling layer. Then, the output of each pooling path is followed by $1x1$ convolutional layer with number of kernels equal to the number of kernels of the previous layer. $1x1$ convolutional layer is used to extract more abundant features. Each $1x1$ convolution layer in the AVG-MAX VPB is followed by a ReLU layer for more stable performance and faster convergence. Then, element-by-element summation is used to combine the extracted feature vectors from ReLU layers. The element-by-element summation is followed by a batch normalization layer and a RELU layer for faster convergence.

## 4. Experimental results

The Database for Mastology Research with Infrared Image (DMR-IR) [45] was developed in 2014 during the PROENG Project at the Institute of Computer Science of the Federal Fluminense University in Brazil. It is currently the only public dataset of breast thermograms. It is used to evaluate the proposed methods in this study. This database is created by collecting the IR images from the Hospital of UFF University and published publicly with the approval of the ethics committee where consent should be signed by any patient. It includes about 5000 thermal images some of them are patients of the hospital and the rest are volunteers. This study used a set of 1000 frontal thermogram images, captured using a FLIR SC-620 IR camera with a resolution of 640×480 pixels from this database (including 500 normal and 500 abnormal

**Table 1. Dataset description.**

| Dataset categories | Training | Validation | Testing | Total |
|---|---|---|---|---|
| Normal | 350 | 75 | 75 | 500 |
| Abnormal | 350 | 75 | 75 | 500 |

subjects). These images contain breasts in various shapes and sizes [6]. The thermal images are resized to a smaller size of 224×224 pixels for faster computation. The dataset is split for segmentation and classification into training, validation and testing sets with the ratio 70:15:15, randomly. The dataset description is included in Table 1.

The proposed models were implemented using the Matlab 2021a platform running on a PC computer system with the following specifications: Intel (R) Core (TM) i7-4770 CPU@3.40GHZ with 64-bit operation system and 16 GB RAM.

## 4. What is section title here?

**4.1. Breast area segmentation.** The thermal image contains unnecessary areas as neck, shoulder, chess, and other parts of the body which acts as noise during the training in CNN models. This phase aims to remove unwanted regions and using the areas predicted to be cancerous as the input to the classification models. Therefore, U-Net network [26] with the concept of VPB and AVG-MAX VPB are used for breast area segmentation from thermal image. According to [26], the original architecture of U-Net has four max-pool layers of size 2*x*2. In this phase, every pooling layer is exchanged by VPB and AVG-MAX VPB. The output of this phase is a binary image as the segmented breast is white and the background is black.

Several evaluation metrics can be used to evaluate the segmentation method [18,46,47] in this paper, global accuracy (Global Acc.), mean accuracy (Mean Acc.), mean of Intersection over Union (Mean IoU) and mean boundary f1 score (Mean BF score) are used to evaluate the breast area segmentation method. The term "mean" refers to the average of the metric of all classes across all images.

**Global accuracy (Global Acc.):** is the ratio of correctly classified pixels, regardless of class, to the total number of pixels. It is used for a quick and computationally inexpensive estimate of the percentage of correctly classified pixels. It is calculated by Eq (9)

$$GlobalAcc. = \frac{T_P + T_N}{T_P + T_N + F_P + F_N} \tag{9}$$

**Accuracy:** indicates the percentage of correctly identified pixels for each class. It is used to know how well each class correctly identifies pixels. It is calculated by Eq (10)

$$Accuracy = \frac{T_P}{T_P + F_N} \tag{10}$$

**Intersection over union (IoU):** is the most common metrics used for segmentation process evaluation. Also, it is known as the Jaccard similarity coefficient. For each class, it is the ratio of correctly classified pixels to the total number of ground truth and predicted pixels in that class. It is used for a statistical accuracy measurement that penalizes false positives. It is calculated by Eq (11)

$$IoU = \frac{T_P}{T_P + F_P + F_N} \tag{11}$$

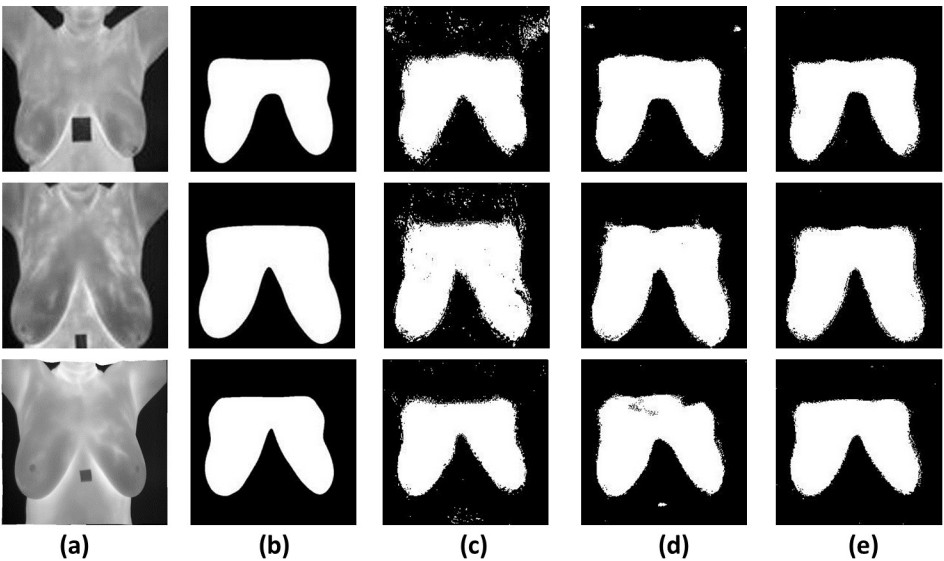

**Fig 6. Examples of segmentation process with U-Net before and after using the proposed pooling blocks.** (a) original images. (b) labels. (c)segmentation with U-Net. (d)segmentation with U-Net+VPB. (e) segmentation with U-Net+ AVG-MAX VPB.

**Boundary f1 Score (BF Score):** indicates how closely the predicted boundary of each class matches the actual boundary. It is used to correlate better with human qualitative assessment than the IoU metric. It is calculated by Eq (12)

$$BF\ Score = \frac{2 * T_P}{2 * T_P + F_P + F_N} \tag{12}$$

Where $T_P$: True Positive, $T_N$: True Negative, $F_P$: False Positive, $F_N$: False Negative.

Adaptive Moment Estimation method (ADAM) [48] is used as optimized algorithm with number of epochs = 30 for the training process of the segmentation method. The initial learning rate for the training process was 1.0e-3. The learning rate used a piecewise schedule and dropped by a factor of 0.30 every 10 epochs, so the network can train rapidly with a high initial learning rate. To preserve memory, the network was trained with a batch of size 8. Fig 6 shows three examples of segmentation process with the original U-Net and U-Net with the proposed pooling methods. The evaluation metrics of the semantic segmentation process with U-net before and after using the VPB and AVG-MAX VPB is shown in Table 2 and Fig 7.

## 4.2. Classification

To show the advantages of the proposed method on the classification process, the vector pooling block and the AVG-MAX pooling block is added to different pretrained CNN models such as ResNet 18 [49], GoogleNet [50] and AlexNet [51] which areused for classification. ResNet

**Table 2. Semantic segmentation evaluation metrics of U-net before and after using VPB and AVG-MAX VPB.**

| Segmentation network | Mean Acc. (%) | Global Acc. (%) | Mean IoU (%) | Mean BFScore (%) |
|---|---|---|---|---|
| U-net | 96.5 | 96.6 | 92.07 | 78.34 |
| U-net with VPB | 97.9 | 98.3 | 95.87 | 88.68 |
| U-net with AVG-MAX VPB | 98.97 | 99.2 | 98.03 | 94.29 |

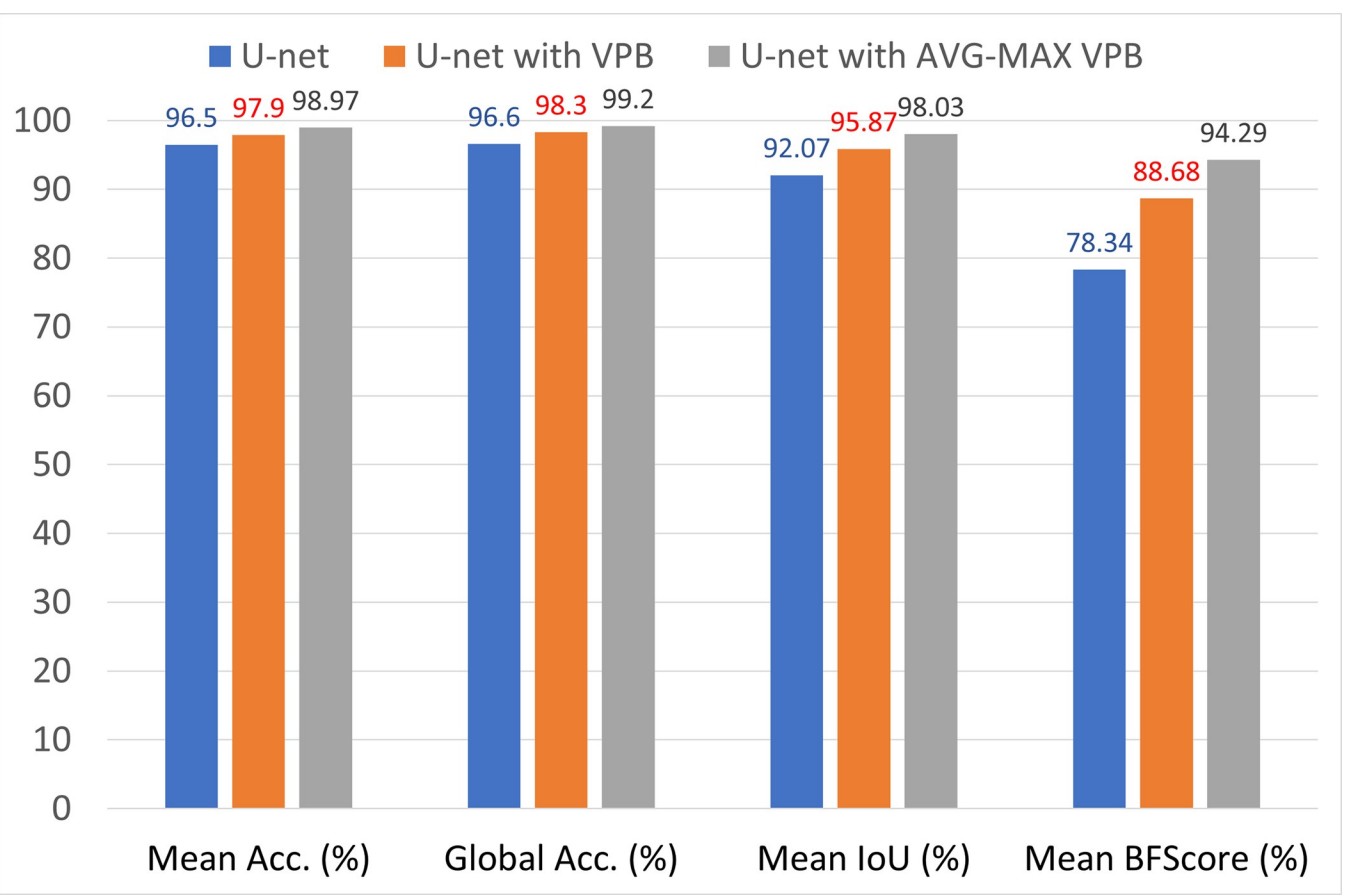

**Fig 7. Illustrative chart for the evaluation metrics of the semantic segmentation process with U-net before and after using the VPB and AVG-MAX VPB presented in Table 2.**

18 architecture has a max-pool layer of size 3*x*3 and an average pool layer of size 7*x*7, Google-Net has four max-pool layers of size 3*x*3 and an average pool layer of size 7*x*7 and AlexNet has three max-pool layers of size 3*x*3. To evaluate the performance of the classification process, classification metrics are used to show how good or bad the classification is.

*Accuracy*: has the same definition of global accuracy. It is used to represent how many instances are completely classified correctly. It is calculated by Eq (9)

*Sensitivity*: Is computed based on how accurately the number of patients with the disease is estimated. It is calculated by Eq (13)

$$Sensitivity = \frac{T_P}{T_P + F_N} \qquad (13)$$

*Specificity*: is calculated based on the number of correctly predicted patients who do not have the disease. It is calculated by Eq (14)

$$Specificity = \frac{T_N}{T_N + F_P} \qquad (14)$$

where, $T_P$: True Positive, $T_N$: True Negative, $F_P$: False Positive, $F_N$: False Negative.

In Table 3, we show the evaluation metrics of the classification process on pretrained CNN networks before and after using the vector pooling block and the AVG-MAX pooling block. In

**Table 3. Evaluation metrics of the classification process on three CNNs before and after using the VPB and the AVG-MAX VPB.**

| CNN model | Accuracy | | | Sensitivity | | | Specificity | | |
|---|---|---|---|---|---|---|---|---|---|
| | Model | Model + VPB | Model + AVG-MAX VPB | Model | Model + VPB | Model + AVG-MAX VPB | Model | Model + VPB | Model + AVG-MAX VPB |
| **AlexNet** | 50 | 90.7 | 99.3 | 0 | 100 | 100 | 100 | 81.3 | 98.7 |
| **GoogleNet** | 79.33 | 96.67 | 100 | 84 | 100 | 100 | 74.67 | 93.3 | 100 |
| **ResNet18** | 93.3 | 100 | 100 | 88 | 100 | 100 | 98.7 | 100 | 100 |

the training process, we use Adaptive Moment Estimation (ADAM)method as solver with batch size of 60 and number of epochs = 30. Also, the training process was started with initial learning rate = 2.0e−3. The training parameters are chosen according to experiments in paper [6]. Figs 8–10 show illustrative charts for the accuracy, sensitivity, and specificity results, respectively for three CNNs before and after using VPB and AVG-MAX VPB presented in Table 3.

## 5. Discussion

In this paper, we present a new design for the pooling layer called vector pooling block (VPB). The vector pooling block consists of two pathways, which focus on extracting features along horizontal and vertical orientation. It makes the CNNs able to collect features on different orientations (horizontal/vertical) by including long and narrow pooling kernels, which is different from the traditional pooling layer, that gathers features from a fixed square kernel. So, it can collect more features that are ignored by the traditional pooling method. Based on the VPB, a pooling module called AVG-MAX VPB is proposed. It can collect informative features by using two types of pooling techniques, Max-pooling and average pooling, with the concept of VPB.

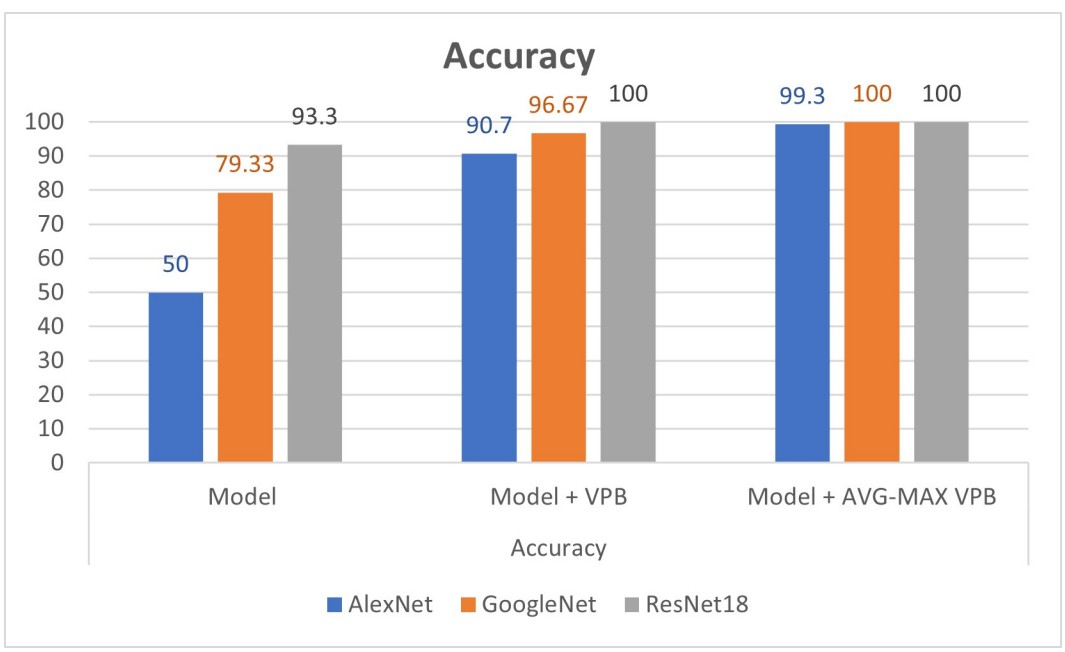

**Fig 8. Illustrative chart for the accuracy results for three CNNs before and after using VPB and AVG-MAX VPB presented in Table 3.**

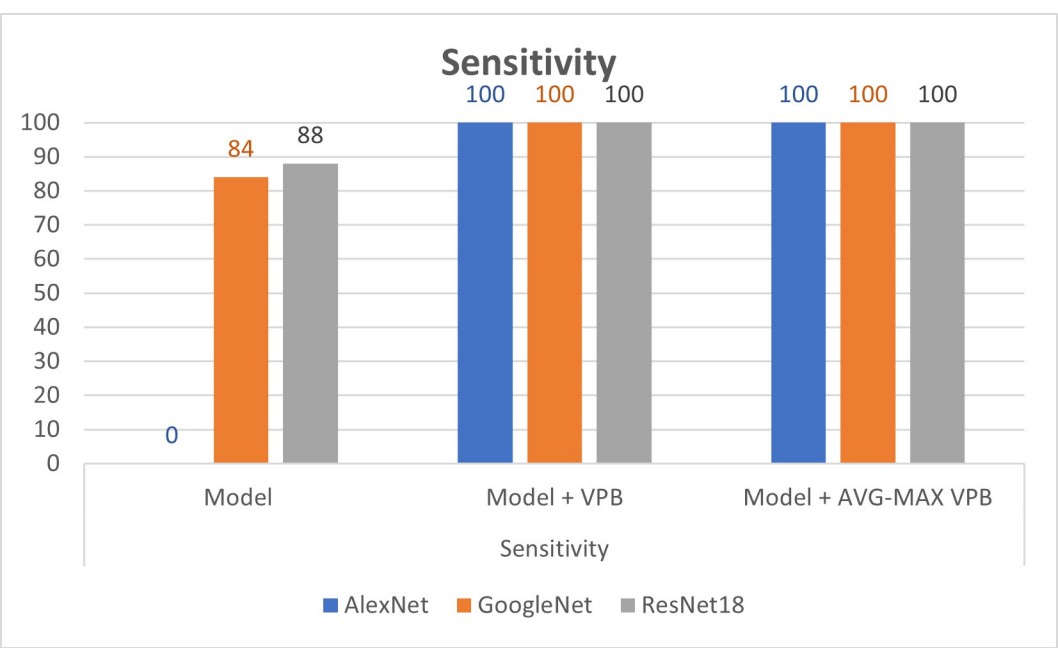

**Fig 9. Illustrative chart for the sensitivity results for three CNNs before and after using VPB and AVG-MAX VPB presented in Table 3.**

The experimental results obtained show our contribution in (1) present a new design for the pooling layer called VPB which focus on extracting features along horizontal and vertical orientation. (2) exchange the pooling layer in CNNs network with the VPB and evaluate its effect in semantic segmentation and classification process. (3) Proposed AVG-MAX VPB

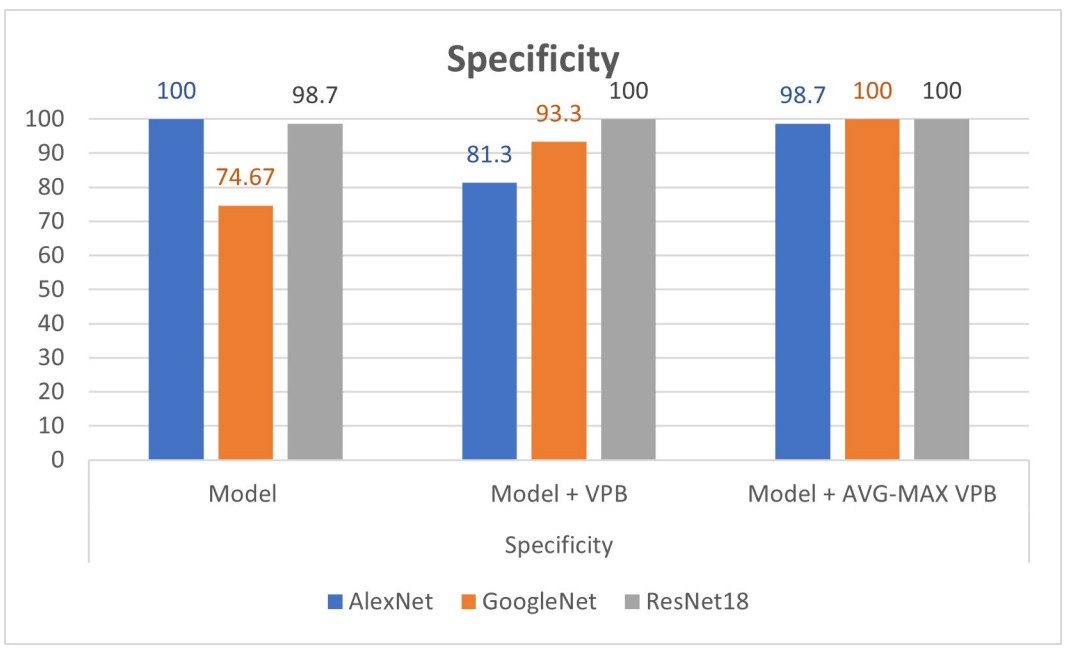

**Fig 10. Illustrative chart for the specificity results for three CNNs before and after using VPB and AVG-MAX VPB presented in Table 3.**

**Table 4. Comparison with other studies on breast cancer detection with CNNs.**

| Ref. | Segmentation method | Classification Method | Results |
|---|---|---|---|
| [25] | FCN | AlexNet | Accuracy = 96.4%, sensitivity = 97.5% and specificity = 97.8% |
| [31] | U-Net | Not defined | Intersection-Over-Union (IoU) = 94.38%. |
| [36] | Connected-UNets | Not defined | Dice Score = 95.88% and IoU = 92.27% |
| [37] | Multi-input CNN | Combining the lateral views and the front view of the thermal images to enhance the performance of the classification model | Accuracy = 97%, specificity = 100% and sensitivity = 83% |
| Proposed method | U-Net with AVG-MAX VPB | AlexNet GoogleNet ResNet-18 | Accuracy = 99.3%, Sensitivity = 100% and specificity = 98.7% with AlexNet Accuracy = 100%, Sensitivity = 100% and specificity = 100% with GoogleNet Accuracy = 100% and Sensitivity = 100% and specificity = 100 with ResNet-18 |

which create a pooling block using average pooling and max pooling based on the concept of vector pooling. (4) plugged AVG-MAX VPB into existing CNN networks evaluate its effect in semantic segmentation and classification process. (5) comparing the proposed models with state-of-the-art models. In Table 2, we study the impact of the VPB and AVG-MAX VPB for breast area extraction from thermal images by plugging them in U-Net network. From Table 2 and Fig 7, we can note that the evaluation metrics of U-Net with the proposed pooling models is better than the evaluation metrics of standard U-Net and the evaluation metrics of U-Net with AVG-MAX VPB is the best. Fig 6 shows three examples of segmentation process with U-Net before and after using the proposed pooling blocks. In Table 3, we study the impact of plugging the VPB and AVG-MAX VPB in pretrained CNNs models such as AlexNet, ResNet18 and GoogleNet for breast tissue classification process from thermal images. From Table 3, Figs 8–10, the evaluation metrics with the usage of the proposed pooling models with pretrained CNNs networks is better than the standard pretrained CNNs and the evaluation metrics of pretrained CNNs with AVG-MAX VPB for the thermal breast tissue classification is the best. To further evaluate our proposed system, as shown in Table 4, a comparison between the proposed system and other studies based on breast area segmentation and breast cancer detection is performed. From this table, we can note that the evaluation metric of our proposed system is better than related work. So, the proposed system outperformed other models.

It is worth mention the computation time of the segmentation and the classification processes with the proposed pooling models is high due to the limitation of the PC capabilities used in this study. But the proposed pooling models are domain-independent, so it can be applied for different computer vision tasks.

## 6. Conclusion

Pooling is a main step in convolutional neural networks that decreases the feature maps' dimensionality, but it has some drawbacks such as losing features and reducing the spatial resolution. In this paper, we present a new design for the pooling layer called vector pooling block (VPB). The VPB consists of two pathways, which focus on extracting features along horizontal and vertical orientation. Based on the vector pooling block, we proposed a pooling module called AVG-MAX VPB. It can collect informative features by using two types of pooling techniques, Max-pooling and average pooling, with the concept of VPB. The VPB and the AVG-MAX VPB are plugged into pretrained CNNs networks such as U-Net, AlexNet, ResNet18 and GoogleNet to show the impact of them in segmentation of the breast area and classification of the breast tissue from thermograms.

Based on the experimental results, the evaluation metrics assured the enhancement of the automatic segmentation of breast area and the classification of breast tissue from thermal images by using the proposed pooling models with CNNs. Furthermore, the proposed pooling models are domain-independent, so it can be applied for different computer vision tasks.

## Supporting information

**S1 Dataset. A sample of the dataset to evaluate the proposed method in this study is uploaded under a file name "S1 Dataset.rar".**
(RAR)

## Acknowledgments

The authors would like to thank the Department of Computer Science and the Hospital of the Federal University Fluminense, Niterói, Brazil, for providing DMR-IR benchmark database which is accessible through an online user-friendly interface (*http://visual.ic.uff.br/dmi*) and used for experiments.

## Author Contributions

**Conceptualization:** Esraa A. Mohamed, Tarek Gaber, Essam A. Rashed.

**Data curation:** Esraa A. Mohamed, Tarek Gaber, Essam A. Rashed.

**Formal analysis:** Esraa A. Mohamed, Tarek Gaber, Essam A. Rashed.

**Investigation:** Esraa A. Mohamed, Tarek Gaber, Essam A. Rashed.

**Methodology:** Esraa A. Mohamed, Essam A. Rashed.

**Project administration:** Esraa A. Mohamed, Tarek Gaber, Omar Karam, Essam A. Rashed.

**Resources:** Esraa A. Mohamed, Tarek Gaber, Essam A. Rashed.

**Software:** Esraa A. Mohamed.

**Supervision:** Tarek Gaber, Omar Karam, Essam A. Rashed.

**Validation:** Esraa A. Mohamed, Tarek Gaber, Essam A. Rashed.

**Visualization:** Esraa A. Mohamed, Tarek Gaber, Essam A. Rashed.

**Writing – original draft:** Esraa A. Mohamed.

**Writing – review & editing:** Tarek Gaber, Omar Karam, Essam A. Rashed.

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
