## [Decision Letter · Decision Letter 0]

5 Sep 2022

PONE-D-22-23310A Novel CCN pooling layer for breast cancer segmentation and classification from thermogramsPLOS ONE

Dear Dr. A. Mohamed,

Thank you for submitting your manuscript to PLOS ONE. After careful consideration, we feel that it has merit but does not fully meet PLOS ONE’s publication criteria as it currently stands. Therefore, we invite you to submit a revised version of the manuscript that addresses the points raised during the review process.

The manuscript should be improved and revised according to the suggestions and comments of the reviewers, specifically focusing on the contextualisation of the study within the state-of-the-art body of knowledge, and improvement of the description and presentation of methodology and results.

We look forward to receiving your revised manuscript.

Kind regards,

Robertas Damaševičius

Academic Editor

PLOS ONE

Journal Requirements:

Additional Editor Comments:

The manuscript should be improved and revised according to the suggestions and comments of the reviewers, specifically focusing on the contextualisation of the study within the state-of-the-art body of knowledge, and improvement of the description and presentation of methodology and results.

Reviewers' comments:

Reviewer's Responses to Questions

**Comments to the Author**

1. Is the manuscript technically sound, and do the data support the conclusions?

Reviewer #1: Partly

Reviewer #2: Yes

2. Has the statistical analysis been performed appropriately and rigorously? 

Reviewer #1: I Don't Know

Reviewer #2: No

3. Have the authors made all data underlying the findings in their manuscript fully available?

Reviewer #1: Yes

Reviewer #2: Yes

4. Is the manuscript presented in an intelligible fashion and written in standard English?

Reviewer #1: Yes

Reviewer #2: Yes

5. Review Comments to the Author

Reviewer #1: Authors should address the following revision:

1) "The most important challenges in breast cancer detection process are accurate segmentation of the breast area and classification of the breast tissue, which play an important role in image guiding surgery, radiological treatment, and clinical computer-assisted diagnosis."- add this ref for the support of this statement: (Breast cancer detection and classification using traditional computer vision techniques: a comprehensive review)

2) Add the importance of deep learning in the domain of medical imaging such as lung cancer, skin cancer, etc. Add the theoratical knowledge with the help of the following references:

- BrainNet: optimal deep learning feature fusion for brain tumor classification

- COVID19 Classification using Chest X-Ray Images: A Framework of CNN-LSTM and Improved Max Value Moth Flame Optimization

- COVID-19 Classification from Chest X-Ray Images: A Framework of Deep Explainable Artificial Intelligence

3) related work should be improved by adding the following works:

- Predicting Breast Cancer Leveraging Supervised Machine Learning Techniques

- Breast Cancer Classification from Ultrasound Images Using Probability-Based Optimal Deep Learning Feature Fusion

4) Add the major contributions under the introduction section.

5) In the methodology section, describe only relevant detail of the proposed method.

6) What is the filter size of pooling layer?

7) What is the nature of output?

8) The detail of datasets should be added in the revised manuscript.

Reviewer #2: The authors propose a strategy for breast cancer diagnosis using thermograms employing pooling layer known as vector pooling block (VPB) which contains two data pathways, focus on extracting features along horizontal and vertical orientations which collect the global and local features. Furthermore, U-Net, AlexNet, ResNet18 and GoogleNet CNN architectures are used for the segmentation and classification. Overall, the work presented in this manuscript is explained well as the authors compare the proposed approach to other existing techniques. Furthermore, the text is clearly written, the methods described clearly, and the results presented in clean figures and easily to understand. Nevertheless, I have some concerns which will improve the quality of the manuscript further. Please see my detailed comments below.

1. The title “CCN pooling layer”. What is meant by CCN?

2. What’s the challenge for diagnosing breast cancer at an early stage compared to advanced cancer from the view of image feature and ML algorithms?

3. Explain the role of recent works of U-NET CNN segmentation as well in your work.

• Maqsood, S., Damasevicius, R., & Shah, F. M. (2021, September). An efficient approach for the detection of brain tumor using fuzzy logic and U-NET CNN classification. In International Conference on Computational Science and Its Applications (pp. 105-118). Springer, Cham.

• Du, G., Cao, X., Liang, J., Chen, X., & Zhan, Y. (2020). Medical image segmentation based on u-net: A review. Journal of Imaging Science and Technology, 64, 1-12.

• Maqsood, S., Damaševičius, R., & Maskeliūnas, R. (2022). TTCNN: A Breast Cancer Detection and Classification towards Computer-Aided Diagnosis Using Digital Mammography in Early Stages. Applied Sciences, 12(7), 3273.

4. Explain the proposed method in more detail. The given information of the proposed method is insufficient. Explain the working of Figures 3,4,5.

5. What is the main motivation behind choosing the selected database? If the images are color or grayscale? Please mention.

6. Describe the computer on which the experiments were performed (OS, CPU, RAM, etc.) and programming environment (language) used to implement the method.

7. The proposed method should also be compared with other methods to show the worth, effectiveness and superiority of the work. The work lacks the discussion section.

8. Add the discussion section in your manuscript and explain how and why your results are superior to other. The information provided in the experimental results portion is insufficient.

9. Patient selection criteria should be provided to show what stage of patients the system is effective for, or if it is effective for any stage of patients.

10. Information on the diagnosing doctor should be included to show whether this accuracy can be obtained by any physician.

11. The computation efficiency of the proposed method should be addressed.

12. There is a need for language improvement. I found some grammatical error texts in the manuscript. The language of the paper needs a review.

6. PLOS authors have the option to publish the peer review history of their article (what does this mean?). If published, this will include your full peer review and any attached files.

Reviewer #1: No

Reviewer #2: No

---

## [Author Response · Author response to Decision Letter 0]

29 Sep 2022

Original Manuscript ID: PONE-D-22-23310 

Original Article Title: “A Novel CCN pooling layer for breast cancer segmentation and classification from thermograms “

To: PLOS ONE

Re: Response to reviewers

Dear Editor,

Thank you for allowing a resubmission of a revised version of the manuscript, with an opportunity to address the reviewers’ comments.

We are uploading (a) our point-by-point response to the comments (below) (Response to Reviewers), (b) a revised manuscript with track changes, and (c) a clean updated manuscript.

Best regards,

Authors, 

Reviewer#1, Concern # 1: "The most important challenges in breast cancer detection process are accurate segmentation of the breast area and classification of the breast tissue, which play an important role in image guiding surgery, radiological treatment, and clinical computer-assisted diagnosis."- add this ref for the support of this statement: (Breast cancer detection and classification using traditional computer vision techniques: a comprehensive review)

Reply: Thank you for your comment. The following reference is added to support this sentence.

 [2] Zahoor S, Lali I U, Khan M A, Javed K, and Mehmood W. Breast Cancer Detection and Classification using Traditional Computer Vision Techniques: A Comprehensive Review. Current Medical Imaging Formerly Current Medical Imaging Reviews. 2021;16(10): 1187–1200. doi: 10.2174/1573405616666200406110547.

Reviewer#1, Concern # 2: Add the importance of deep learning in the domain of medical imaging such as lung cancer, skin cancer, etc. Add the theoratical knowledge with the help of the following references:

- BrainNet: optimal deep learning feature fusion for brain tumor classification

- COVID19 Classification using Chest X-Ray Images: A Framework of CNN-LSTM and Improved Max Value Moth Flame Optimization

- COVID-19 Classification from Chest X-Ray Images: A Framework of Deep Explainable Artificial Intelligence

Reply: Thank you for your comment, the manuscript is revised as suggested with the following sentences (See Page 2 in Manuscript) and the following references are added.

Deep learning is a machine learning technology that uses multilayer convolutional neural networks (CNNs) [11]. It has a significant effect in fields associated to medical imaging such as brain tumor detection [12] which proposes a fully automated design to classify brain tumors, COVID-19 as in [13] which propose a deep learning and explainable AI technique for the diagnosis and classification of COVID-19 using chest X-ray images and [14] which proposed a CNN-LSTM and improved max value features optimization framework for COVID-19 classification to address the issue of multisource fusion and redundant features, lung cancer [15], which developed and validated a deep learning-based model using the segmentation method and assessed its ability to detect lung cancer on chest radiographs, etc.

[12] Zahid U et al. BrainNet: Optimal Deep Learning Feature Fusion for Brain Tumor Classification. Comput Intell Neurosci. 2022: 1–13. doi: 10.1155/2022/1465173.

[13] Khan M A et al. COVID-19 Classification from Chest X-Ray Images: A Framework of Deep Explainable Artificial Intelligence. Comput Intell Neurosci. 2022; 1–14. doi: 10.1155/2022/4254631.

[14] Hamza A et al. COVID-19 classification using chest X-ray images: A framework of CNN-LSTM and improved max value moth flame optimization. Front Public Health. 2022; 10. doi: 10.3389/fpubh.2022.948205.

[15] Shimazaki A et al. Deep learning-based algorithm for lung cancer detection on chest radiographs using the segmentation method. Sci Rep. 2022; 12(1):727. doi: 10.1038/s41598-021-04667-w.

Reviewer#1, Concern # 3: related work should be improved by adding the following works:

- Predicting Breast Cancer Leveraging Supervised Machine Learning Techniques

- Breast Cancer Classification from Ultrasound Images Using Probability-Based Optimal Deep Learning Feature Fusion

Reply: Thank you for your comment, the manuscript is revised as suggested (see page 2) and the following references are added

[3] Jabeen K et al. Breast Cancer Classification from Ultrasound Images Using Probability-Based Optimal Deep Learning Feature Fusion. Sensors. 2022; 22(3): 807. doi: 10.3390/s22030807.

 [7] Aamir S et al. Predicting Breast Cancer Leveraging Supervised Machine Learning Techniques. Comput. Math Methods Med. 2022; 1–13. doi: 10.1155/2022/5869529.

The sentences that contain these references are copied below:

Several breast imaging modalities are currently being used for early detection of breast cancer such as ultrasound [3], mammography [4], MRI[5], thermography [6][7], etc. Computer aided detection (CAD) system is used for the diagnosis of breast cancer. This diagnosis contains several methods and techniques, including image processing, machine learning [7], data analysis, artificial intelligence [8], and deep learning [6], [9], [10].

Reviewer#1, Concern # 4: Add the major contributions under the introduction section.

Reply: The major contributions are already existed under the Introduction Section at page 4 in Manuscript. They have been revisited and improved. Also, it is copied below:

The major contributions of this paper are as following:

 Proposing a novel design for the pooling layer called VPB, which focus on extracting features along horizontal and vertical orientations and makes the CNNs able to collect both global and local features by including long and narrow pooling kernels.

 Proposing another new pooling block based on the VPB called AVG-MAX VPB, which creates a pooling block using average pooling and max pooling based on the concept of vector pooling. 

 Proposing an enhanced CNN (VPB-CNN) by embedding the proposed pooling models above and then used it in the semantic segmentation and classification of thermography breast cancer problem.

 Conducing a thorough evaluation of the enhanced CNN (VPB-CNN) including the standard networks such as U-Net, AlexNet, ResNet18 and GoogleNet. This showed that the VPB-CNN outperformed these standard ones. 

Reviewer#1, Concern # 5: In the methodology section, describe only relevant detail of the proposed method.

Reply: Thank you for your comment, the manuscript is revised as suggested (See Pages 7,8&9 in Manuscript).

Reviewer#1, Concern # 6: What is the filter size of pooling layer?

Reply: The proposed method can be applied on different filter size of the pooling layer. The following sentences are added to the Manuscript to confirm our reply.

In Page 9, we add

Therefore, U-Net network [26] with the concept of VPB and AVG-MAX VPB are used for breast area segmentation from thermal image. According to [26], the original architecture of U-Net has four max-pool layers of size 2x2. 

In Page 12, we add

To show the advantages of the proposed method on the classification process, the vector pooling block and the AVG-MAX pooling block is added to different pretrained CNN models such as ResNet 18[49], GoogleNet [50]and AlexNet [51] which areused for classification. ResNet 18 architecture has a max-pool layer of size 3x3 and an average pool layer of size 7x7, GoogleNet has four max-pool layers of size 3x3 and an average pool layer of size 7x7 and AlexNet has three max-pool layers of size 3x3.

Reviewer#1, Concern # 7: What is the nature of output?

Reply: The output of the segmentation network is a binary image as the segmented breast is white and the background is black as in Fig 6. The output of the classification networks is a digit which represents the reference to class category. 

Reviewer#1, Concern # 8: The detail of datasets should be added in the revised manuscript.

Reply: Thank you for your comment, the manuscript is revised as suggested (See Page 9) and is copied below 

The Database for Mastology Research with Infrared Image (DMR-IR) [41] was developed in 2014 during the PROENG Project at the Institute of Computer Science of the Federal Fluminense University in Brazil. It is currently the only public dataset of breast thermograms It is used to evaluate the proposed methods in this paper. This database is created by collecting the IR images from the Hospital of UFF University and published publicly with the approval of the ethics committee where consent should be signed by any patient. It includes about 5000 thermal images some of them are patients of the hospital and the rest are volunteers. This paperused a set of 1000 frontal thermogram images, captured using a FLIR SC-620 IR camera with a resolution of 640×480 pixels from this database (including 500 normal and 500 abnormal subjects). These images contain breasts in various shapes and sizes [6]. The dataset is split for segmentation and classification into training, validation and testing sets with the ratio 70:15:15, randomly. The dataset description is included in Table1.

Reviewer#2, Concern # 1: The title “CCN pooling layer”. What is meant by CCN?

Reply: We would like to apologize for this fetal error. We change it to CNN.

Reviewer#2, Concern # 2: What’s the challenge for diagnosing breast cancer at an early stage compared to advanced cancer from the view of image feature and ML algorithms?

Reply: Thank you for addressing this important point. In early stages of breast cancer, the detectability of abnormal tissues is challenging using standard approaches such as mammography as it localized in small region and usually presented in texture similar to surrounding normal breast tissues [16]. However, in thermography, the detectability is based on the change in body temperature which open additional potentials for early detection using different physical features. Machine learning and deep learning approaches lead to improve the performance of detectability due to the ability to recognize the image features without the need of feature engineering.

[16] Houssami N, Given-Wilson R, and Ciatto S. Early detection of breast cancer: Overview of the evidence on computer-aided detection in mammography screening. J Med Imaging Radiat Oncol. 2009; 53(2): 171–176. doi: 10.1111/j.1754-9485.2009.02062.x.

Reviewer#2, Concern # 3: Explain the role of recent works of U-NET CNN segmentation as well in your work.

• Maqsood, S., Damasevicius, R., & Shah, F. M. An efficient approach for the detection of brain tumor using fuzzy logic and U-NET CNN classification. In International Conference on Computational Science and Its Applications. 2021; 105-118. Springer, Cham.

• Du, G., Cao, X., Liang, J., Chen, X., & Zhan, Y. Medical image segmentation based on u-net: A review. Journal of Imaging Science and Technology.2020; 64: 1-12.

• Maqsood, S., Damaševičius, R., & Maskeliūnas, R. TTCNN: A Breast Cancer Detection and Classification towards Computer-Aided Diagnosis Using Digital Mammography in Early Stages. Applied Sciences, 2022; 12(7): 3273.

Reply: Thank you for your comment, the manuscript is revised as suggested (See Page 2&4 in Manuscript) and the following references are added:

[17] Maqsood S, Damaševičius R, & Maskeliūnas R. TTCNN: A Breast Cancer Detection and Classification towards Computer-Aided Diagnosis Using Digital Mammography in Early Stages. Applied Sciences.2022; 12(7): 3273. doi: 10.3390/app12073273.

 [27] Maqsood S, Damasevicius R, & Shah F M. An efficient approach for the detection of brain tumor using fuzzy logic and U-NET CNN classification. In International Conference on Computational Science and Its Applications. 2021: 105-118. 

[28] Du G, Cao X, Liang J, Chen X, and Zhan Y. Medical image segmentation based on u-net: A review. Journal of Imaging Science and Technology. 2020; 64: 1-12., doi: 10.2352/J.ImagingSci.Technol.2020.64.2.020508.

Sentences that contain these references are copied below.

In Page 2, we add:

Therefore, CNNs have been successfully applied for the diagnosis of breast cancer in recent years [10], [6], [17].

In Page 4, we add:

U-Net is improved and extended from FCN [26][26]. It can be used for classification such as [27] which uses U-Net CNN for the classification of brain tumor. It is widely applied to several medical image segmentation tasks [28] such as lung [29], skin lesions [30], etc. Also, it is used for breast area segmentation from thermal images.

Reviewer#2, Concern # 4: Explain the proposed method in more detail. The given information of the proposed method is insufficient. Explain the working of Figures 3,4,5.

Reply: Thank you for your comment, the manuscript is revised as suggested (See Page 8 in Manuscript)

.

Reviewer#2, Concern # 5: What is the main motivation behind choosing the selected database? If the images are color or grayscale? Please mention.

Reply: Up to the best of author’s knowledge, this database is currently the only public dataset of breast thermograms. Also, it is the most dataset used for the diagnosis of breast cancer from thermograms in recent years.

The images are grayscale.

Reviewer#2, Concern # 6: Describe the computer on which the experiments were performed (OS, CPU, RAM, etc.) and programming environment (language) used to implement the method.

Reply: Thank you for your comment, the manuscript is revised as suggested (See Page 9 in Manuscript). Also, it is copied below:

The proposed models was implemented using the Matlab 2021a platform running on a PC computer system with the following specifications: Intel (R) Core (TM) i7-4770 CPU@3.40GHZ with 64-bit operation system and 16 GB RAM.

Reviewer#2, Concern # 7: The proposed method should also be compared with other methods to show the worth, effectiveness and superiority of the work. The work lacks the discussion section.

Reply: The manuscript is revised as suggested by the reviewer (See Pages 15 in Manuscript) and is shown in Table4

Ref. Segmentation method Classification Method Results 

[25]

FCN AlexNet Accuracy=96.4%, sensitivity=97.5% and specificity=97.8%

[31]

U-Net Not defined Intersection-Over-Union (IoU)=94.38%.

[36]

Connected-UNets Not defined Dice Score=95.88% and IoU=92.27%

[37]

Multi-input CNN combining the lateral views and the front view of the thermal images to enhance the performance of the classification model Accuracy=97%, specificity=100% and sensitivity= 83%

Proposed method U-Net with AVG-MAX VPB AlexNet

GoogleNet

ResNet-18 Accuracy=99.3, Sensitivity = 100% and specificity=98.7 with AlexNet

Accuracy=100%, Sensitivity = 100% and specificity=100 with GoogleNet

Accuracy=100% and Sensitivity = 100% and specificity= 100 with ResNet-18

Reviewer#2, Concern # 8: Add the discussion section in your manuscript and explain how and why your results are superior to other. The information provided in the experimental results portion is insufficient.

Reply: The Discussion Section is already existed in the Manuscript in page 13, 14&15 after the Experimental Results Section. It has been revisited and improved with a comparison between the proposed models and other studies on breast cancer detection with CNNs in Table 4.

Reviewer#2, Concern # 9: Patient selection criteria should be provided to show what stage of patients the system is effective for, or if it is effective for any stage of patients.

Reply: Thank you for addressing this important point. The database used in this study is divided into two categories (healthy or sick). It doesn’t contain information about breast cancer stages of sick patients. This paper used a set of 1000 frontal thermogram images from this database (including 500 healthy and 500 sick patients). The dataset is split for segmentation and classification into training, validation and testing sets with the ratio 70:15:15, randomly. The dataset description is included in Table1. In general, thermography is better than mammography in detecting the breast cancer in its early stages, and this is the most important thing in diagnosing the cancer.

Reviewer#2, Concern # 10: Information on the diagnosing doctor should be included to show whether this accuracy can be obtained by any physician.

Reply: Details on the dataset and diagnosis results are described in the original publication of the data set in references [45]. Considerations regarding the diagnosis quality and experience of the physicians is out of the scope of the current manuscript.

Reviewer#2, Concern # 11: The computation efficiency of the proposed method should be addressed.

Reply: Thank you for your comment. In this paper, global accuracy (Global Acc.), Mean Accuracy (Mean Acc.), Mean of Intersection over Union (Mean IoU) and Mean Boundary f1 score (Mean BF score) are used to evaluate the breast area segmentation method. Global accuracy (Global Acc.) is used for a quick and computationally inexpensive estimate of the percentage of correctly classified pixels, Mean Accuracy (Mean Acc.) is used to know how well each class correctly identifies pixels, Mean of Intersection over Union (Mean IoU) is used for a statistical accuracy measurement that penalizes false positives, and Mean Boundary f1 Score (Mean BF score) is used to correlate better with human qualitative assessment than the IoU metric. Also, Accuracy, Sensitivity and Specificity are used to evaluate the performance of the classification process. Accuracy is used to represent how many instances are completely classified correctly. Sensitivity is used to indicate how many patients have the disease are correctly estimated. Specificity is used to indicate how many patients do not have the disease are predicted right. In addition, a Comparison between the proposed models and other studies on breast cancer detection with CNNs is performed in Table 4 in the Discussion Section.

Reviewer#2, Concern # 12: There is a need for language improvement. I found some grammatical error texts in the manuscript. The language of the paper needs a review.

Reply. Thank you for your comment, the manuscript is carefully revised for language issues.

---

## [Decision Letter · Decision Letter 1]

10 Oct 2022

A Novel CNN pooling layer for breast cancer segmentation and classification from thermograms

PONE-D-22-23310R1

Dear Dr. Gaber,

We’re pleased to inform you that your manuscript has been judged scientifically suitable for publication and will be formally accepted for publication once it meets all outstanding technical requirements.

Kind regards,

Robertas Damaševičius

Academic Editor

PLOS ONE

Additional Editor Comments (optional):

Reviewers' comments:

Reviewer's Responses to Questions

**Comments to the Author**

1. If the authors have adequately addressed your comments raised in a previous round of review and you feel that this manuscript is now acceptable for publication, you may indicate that here to bypass the “Comments to the Author” section, enter your conflict of interest statement in the “Confidential to Editor” section, and submit your "Accept" recommendation.

Reviewer #1: (No Response)

Reviewer #2: All comments have been addressed

2. Is the manuscript technically sound, and do the data support the conclusions?

Reviewer #1: (No Response)

Reviewer #2: Yes

3. Has the statistical analysis been performed appropriately and rigorously? 

Reviewer #1: (No Response)

Reviewer #2: Yes

4. Have the authors made all data underlying the findings in their manuscript fully available?

Reviewer #1: (No Response)

Reviewer #2: Yes

5. Is the manuscript presented in an intelligible fashion and written in standard English?

Reviewer #1: (No Response)

Reviewer #2: Yes

6. Review Comments to the Author

Reviewer #1: Authors well revised this manuscript and it can be accepted in the current form. Also, the references section is improved.

Reviewer #2: Authors attended correctly to all of my suggestions. So, I am satisfied with the revised version. The revision is acceptable.

7. PLOS authors have the option to publish the peer review history of their article (what does this mean?). If published, this will include your full peer review and any attached files.

Reviewer #1: No

Reviewer #2: No

---

## [Editor Report · Acceptance letter]

12 Oct 2022

PONE-D-22-23310R1 

A Novel CNN pooling layer for breast cancer segmentation and classification from thermograms 

Dear Dr. Gaber:

I'm pleased to inform you that your manuscript has been deemed suitable for publication in PLOS ONE. Congratulations! Your manuscript is now with our production department. 

Kind regards, 

on behalf of

Professor Robertas Damaševičius 

Academic Editor

PLOS ONE